Genome-wide identification and characterization of aquaporin gene family in Beta vulgaris

Kong Weilong 1
Yang Shaozong 1
Wang Yulu 1
Bendahmane Mohammed 2
Fu Xiaopeng fuxiaopeng@mail.hzau.edu.cn 1
1 College of Horticulture and Forestry Sciences, Huazhong Agricultural University, Key Laboratory of Horticultural Plant Biology, Ministry of Education , Wuhan, Hubei , China
2 INRA-CNRS-Lyon1-ENS, Laboratoire Reproduction et Developpement des Plantes , Ecole Normale Supérieure Lyon , France
Hyten David
Electronic publication date: 2017 Sep 19
Publication date: 2017
Volume: 5
Electronic Location ID: e3747
Received 2017 Apr 13; Accepted 2017 Aug 8
Copyright: ©2017 Kong et al.
Copyright year: 2017
Copyright holder: Kong et al.
License: This is an open access article distributed under the terms of the Creative Commons Attribution License, which permits unrestricted use, distribution, reproduction and adaptation in any medium and for any purpose provided that it is properly attributed. For attribution, the original author(s), title, publication source (PeerJ) and either DOI or URL of the article must be cited.
License URL: https://creativecommons.org/licenses/by/4.0/

Keywords: Gene structure, Expression profile, Beta vulgaris, Aquaporins, Abiotic stress

Funding: National Natural Science Foundation of China 31000918 Fundamental Research Funds for the Central Universities 2662015PY052 2662016PY041 This study was supported by the National Natural Science Foundation of China (31000918) and the Fundamental Research Funds for the Central Universities (2662015PY052 and 2662016PY041). The funders had no role in study design, data collection and analysis, decision to publish, or preparation of the manuscript.

==============================
Aquaporins (AQPs) are essential channel proteins that execute multi-functions throughout plant growth and development, including water transport, uncharged solutes uptake, stress response, and so on. Here, we report the first genome-wide identification and characterization AQP (BvAQP) genes in sugar beet (Beta vulgaris), an important crop widely cultivated for feed, for sugar production and for bioethanol production. Twenty-eight sugar beet AQPs (BvAQPs) were identified and assigned into five subfamilies based on phylogenetic analyses: seven of plasma membrane (PIPs), eight of tonoplast (TIPs), nine of NOD26-like (NIPs), three of small basic (SIPs), and one of x-intrinsic proteins (XIPs). BvAQP genes unevenly mapped on all chromosomes, except on chromosome 4. Gene structure and motifs analyses revealed that BvAQP have conserved exon-intron organization and that they exhibit conserved motifs within each subfamily. Prediction of BvAQPs functions, based on key protein domains conservation, showed a remarkable difference in substrate specificity among the five subfamilies. Analyses of BvAQPs expression, by mean of RNA-seq, in different plant organs and in response to various abiotic stresses revealed that they were ubiquitously expressed and that their expression was induced by heat and salt stresses. These results provide a reference base to address further the function of sugar beet aquaporins and to explore future applications for plants growth and development improvements as well as in response to environmental stresses.

Introduction

AQPs are known to facilitate water transport and other small nutrients through cell membranes (Maurel et al., 2009; Maurel et al., 2008). Since the discovery of the first aquaporin (AQP1) in mammals, AQPs were identified in many microorganisms, plants and animals (Gomes et al., 2009).

Plant aquaporin families are complex, and are composed of a large number of genes. For example, there are 35 AQPs in Arabidopsis thaliana, 31 in Zea mays, 34 in Oryza sativa, 55 in Populus trichocarpa and 66 AQPs in Glycine max (Johanson & Kjellbom, 2001; Nguyen, Moon & Jung, 2013; Chaumont et al., 2001; Gupta & R, 2009; Zhang et al., 2013). In plants AQPs play major roles in water and solute transport, in maintaining water homeostasis and in the response to environment stresses. AQPs roles in glycerol, urea, boric acid, silicic acid, H2O2, NH3 and CO2 transport through cell membranes were reported to be important for cytoplasm homeostasis, seed germination, embolism recovery, petal and leaf movement, guard cells closure, fruit ripening and maintenance of cell turgor under various stresses (Fitzpatrick & Reid, 2009; Maurel et al., 2009; Maurel et al., 2008; Mitani-Ueno et al., 2011; Heinen & Chaumont, 2009; Maurel et al., 2008; Prado & Maurel, 2013; Uehlein & Kaldenhoff, 2008; Wudick & Maurel, 2009).

To date, AQPs are recognized into seven subfamilies: PIPs, TIPs, NIPs, SIPs, XIPs, GIPs, and HIPs (Anderberg, Kjellbom & Johanson, 2012; Danielson & Johanson, 2008). Green plants usually contain four subfamilies: PIPs, TIPs, NIPs, and SIPs. However, members of XIPs subfamily were also found in some dicots, such as Solanum lycopersicum (Venkatesh, Yu & Park, 2013), Populus trichocarpa (Gupta & Sankararamakrishnan, 2009) and Glycine max (Cheng et al., 2013), but were absent in Brassicaceae and monocots (Danielson & Johanson, 2008). GIPs and HIPs subfamilies were reported in moss (Physcomitrella patens) and fern (Selaginella moellendorffii).

AQPs are highly conserved in all living organisms, consisting of six transmembrane domains (TM1–TM6) connected by five loops (LA-LE). NPA motifs, the ar/R selectivity and Froger’s position are critical for AQPs functions. Two NPA motifs (Asn-Pro-Ala) are located on LB and LE, forming a central aqueous pore in the middle of the lipid bilayer involved in proton exclusion and substrate selectivity (Bansal & Sankararamakrishnan, 2007). The ar/R selectivity is formed by four resides from TM2 (H2), TM5 (H5), LE (LE1 and LE2) and acts as a size-exclusion barrier (Hove & Bhave, 2011; Mitani-Ueno et al., 2011). Froger’s position consists of five conserved resides (P1–P5) and discriminations between AQP-type and GIP-type AQPs (Froger et al., 1998). Nine specificity-determining positions for non-aqua substrates (i.e., urea, boric, acid, silicic, ammonia, carbon dioxide and hydrogen peroxide (H2O2)) were also proposed based on a comprehensive analysis of functionally characterized AQPs (Hove & Bhave, 2011).

Sugar beet belongs to Caryophyllales, which lays on basal taxa of core dicots. It is an important crop in temperate climates region and provides nearly 30% of the world’s annual sugar production (Dohm et al., 2014). It is also used as a source for animal feed and for bioethanol production.

To date, little is known about AQPs in sugar beet and in Caryophyllales. So far, only information on 26 AQPs, grouped in five subfamilies (eight PIPs, 11 TIPs, four NIPs, two SIPs and one XIP), was reported in carnation (Morita et al., 2017). More information on AQPs in Caryophyllales plants is therefore required to help understand their function and evolution.

Here we used the available high-quality genome sequence (Dohm et al., 2014) and RNA-seq datasets (Minoche et al., 2015) of sugar beet to identify and characterize the expression of BvAQPs. We report the distribution of BvAQPs on chromosomes, phylogeny analysis, gene structure, subcellular location, conserved resides and conserved elements. We propose putative functions of sugar beet AQPs based on their detailed genes expression patterns analysis by means of RNA-seq.

Methods

Identification and distribution of AQP genes in sugar beet

BvAQPs were identified by HMM (Hidden Markov Model) and BLAST homology search. The sugar beet predicted proteome was collected using the sugar beet genome RefBeet-1.2 (http://bvseq.molgen.mpg.de/Genome/Download/index.shtml). The Hidden Markov model (HMM) of the MIP domain (PF00230) was downloaded from the Sanger database (http://pfam.xfam.org/family/PF00230). PF00230 was then used to query the predicted Sugar beet proteome using HMMER 3.0 software (http://hmmer.org/). 35 Arabidopsis AtAQPs were download from TAIR Database (https://www.arabidopsis.org/browse/genefamily/Aquaporins.jsp) and then used to search BvAQPs with BLASTp tool using NCBI sugar beet genome (https://www.ncbi.nlm.nih.gov/genome/?term=Beta+vulgaris) and sugar beet genome (http://www.genomforschung.uni-bielefeld.de/en/projects/annobeet) (Baranwal, Negi & Khurana, 2016) with cut-off E-value of e−5. All no-redundant gene sequences were analyzed by SMART (http://smart.embl-heidelberg.de/) and Pfam (http://pfam.xfam.org/search/sequence). Sequences encoding complete MIP domain and two NPA motifs were considered as putative AQP genes.

Additionally, the molecular weight (MW) and isoelectric point (PI) of BvAQPs were calculated by ExPASy (http://www.expasy.org/); transmembrane helical domains (TMHs) were assessed by TMHMM Sever v.2.0 (http://www.cbs.dtu.dk/services/TMHMM/); the subcellular localization of BvAQPs were predicted using Plant-mPLoc (http://www.csbio.sjtu.edu.cn/bioinf/plant-multi/) and WolF PSORT (http://www.genscript.com/wolf-psort.html). The position of the AQP genes on the sugar beet chromosomes were identified based on position information from the sugar beet genome database and the distribution graph of AQP genes was drawn by MapInspect software (http://mapinspect.software.informer.com/).

Phylogenetic analyses and sequence alignments

BvAQPs were aligned with AtAQPs from Arabidopsis (https://www.arabidopsis.org/) DcAQPs from Dianthus caryophyllus (Morita et al., 2017; http://carnation.kazusa.or.jp/blast.html), by using Clustal W. Phylogenetic tree was built by MEGA6.0 (http://www.megasoftware.net/history.php) using the neighbor-joining (NJ) method, with 1,000 times bootstrap replicates. BvAQPs were named based on their sequence homology and phylogenetic analyses. Greek letters (α, β) were used to denote the transcripts derived from the same gene.

BvAQPs and two S. tuberosum AQPs (Venkatesh, Yu & Park, 2013) were aligned by DNAMAN (http://dnaman.software.informer.com/) with default parameters. Two NPA motifs, ar/R selectivity filter and Froger’s position were inferred from the multiple sequence alignment result from DNAMAN. Specificity-determining positions (SDP1-SDP9) from alignments with the structure resolved Spinacia oleracea PIP2;1 and functionally characterized AQPs as being collected by Hove and Bhave (Hove & Bhave, 2011).

Exon-intron structure, tandem duplication events and conserved motifs distribution

The exon-intron structure was performed by GSDS 2.0 (http://gsds.cbi.pku.edu.cn/) based on genes coding sequences and on the annotated genome. Tandem duplication events were analyzed in BvAQP genes (Gu et al., 2002; He et al., 2012). Conserved motifs of BvAQPs were analyzed by MEME suit (http://meme-suite.org/), and the parameters were set as follows: maximum number is 10, other default parameters.

Expression analysis of sugar beet AQP genes

RNA-seq data (SRX287608–SRX287615) were collected from NCBI and used to analyse the expression profiles of AQP genes in different organs and tissues (seedling, root, leaf, inflorescence and seed). Salt and heat tolerance is largely dependent on the plant ability to maintain optimal water status in leaves. The adjustment of water relation under salinity involves changes in the transcriptional activity of genes encoding AQPs. Expression variations of BvAQPs in young leaf under salt or heat stress, were analyzed using RNA-seq data (SRX647324; SRX647712; SRX647714) (Minoche et al., 2015).

HISAT2 was used to align raw reads to the reference genome, and StringTie was used to calculate gene expression (Stracke et al., 2014). The heat map for tissue-specific expression profile was generated based on the Log2RPKM values for each gene in all the tissue samples using R package (Gentleman et al., 2004). Stress inducible profile of BvAQP genes in young leaf, RPKM values were normalized to untreated controls, and expression fold changes in genes were shown in terms of Log2fold (Venkatesh, Yu & Park, 2013).

Results

Identification, classification and properties of BvAQP genes in sugar beet

In total 28 non-redundant genes were assigned as putative AQPs. The predicted protein sequence of all AQPs ranged from 236 to 327 amino acids (Table 1). Sequence cluster analysis of AQPs from A. thaliana, D. caryophyllus and sugar beet permitted to group them into five subfamilies: seven PIPs, eight TIPs, nine NIPs, three SIPs and one XIPs (Fig. 1). PIPs subfamily was further divided into three PIP1s and four PIP2s subgroups. The TIPs subfamily included five subgroups (three TIP1s, two TIP2s, one TIP3, one TIP4, one TIP5). NIPs subfamily was dived into five subgroups (one NIP1, two NIP4s, two NIP5s, three NIP6s, one NIP7). SIPs subfamily divided into two subgroups (two SIP1s, one SIP2). Only one member (XIP1) composed XIPs subfamily. A pair of transcripts were found in SIP1 subgroup, named BvSIP1;1 α and SIP1;1 β.

Table 1 Identification of BvAQP genes using sugar beet genome data.

Family	Gene	Gene ID	Gene code	Protein length (aa)	MW (kDa)	pI	TMHs	Plant-mPLoc	WoLF PSORT	
PIP	BvPIP1;1	fpur	Bv1g004510_fpur.t1	289	31.1	8.84	5	plas	cyto	
BvPIP1;2	shpz	Bv1g004520_shpz.t1	285	30.67	9.05	6	plas	plas	
BvPIP1;3	gqok	Bv2g024120_gqok.t1	286	30.74	9.14	6	plas	plas	
BvPIP2;1	cqnr	Bv7g163390_cqnr.t1	284	30.32	8.31	6	plas	plas	
BvPIP2;2	ixem	Bv9g210030_ixem.t1	288	31.09	7.08	6	plas	plas	
BvPIP2;3	yige	Bv9g210020_yige.t1 (XP_010689549.1)a	274	29.64	8.97	6	plas	plas	
BvPIP2;4	reke	Bv9g216070_reke.t1	281	30.13	8.84	6	plas	plas	
TIP	BvTIP1;1	kzkq	Bv7ug180930_kzkq.t1	254	26.06	5.38	6	vacu	plas	
BvTIP1;2	iuuk	Bv2g037380_iuuk.t1	252	26.3	5.92	6	vacu	vacu	
BvTIP1;3	ynzf	Bv7g176430_ynzf.t1	248	25.46	5.13	6	vacu	vacu	
BvTIP2;1	dkzm	Bv9g223310_dkzm.t1	247	25.26	5.6	6	vacu	plas	
BvTIP2;2	xunf	Bv5g104980_xunf.t1	249	25.09	4.7	7	vacu	vacu	
BvTIP3;1	dreg	Bv8g190600_dreg.t1	257	27.13	7.07	5	vacu	nucl/mito/vacu	
BvTIP4;1	ydno	Bv2g032200_ydno.t1	247	26.1	6.57	6	vacu	cyto/vacu	
BvTIP5;1	gghp	Bv3ug068240_gghp.t1	255	26.54	8.47	6	plas	chlo	
NIP	BvNIP1;1	ughi	Bv8ug202570_ughi.t1	292	30.87	8.91	6	plas	plas	
BvNIP4;1	aejh	Bv2g027680_aejh.t1	273	29.3	8.87	6	plas	vacu	
BvNIP4;2	xash	Bv2g027660_xash.t1	299	32.62	8.27	6	plas	plas	
BvNIP5;1	gkiq	Bv6TE021760_gkiq.t1 (XP_010680949.1)a	261	27.61	8.96	6	plas	vacu	
BvNIP5;2	oani	Bv6g139140_oani.t1	298	30.9	8.73	6	plas	plas	
BvNIP6;1	hmzo	Bv9g225280_hmzo.t1	306	31.75	7.66	6	plas	plas	
BvNIP6;2	zkgo	Bv5g108450_zkgo.t1 (XP_010677474.1)a	266	27.46	7.82	6	plas	plas	
BvNIP6;3	jecw	Bv5g108440_jecw.t1 (XP_010677699.1)a	327	35.08	9.03	6	plas	plas	
BvNIP7;1	kqew	Bv3ug070540_kqew.t1	289	30.72	7.13	7	plas	plas	
SIP	BvSIP1;1 α	zywx	Bv2g035790_zywx.t1 (XP_010669579.1)a	250	26.46	9.46	5	plas	plas	
BvSIP1;1 β	fzwq	Bv2g035780_fzwq.t1	247	26.39	9.74	6	plas/vacu	vacu	
BvSIP2;1	qzqg	Bv3g064810_qzqg.t1 (XP_010673209.1)a	236	26.02	9.56	5	plas	vacu	
XIP	BvXIP1;1	iwpe	Bv9g217040_iwpe.t1	312	34.13	8.38	7	plas	plas	
Notes.

Note1: Best possible cell localization prediction by the Plant-mPLoc and WoLF PSORT tool (Chlo: chloroplast; Cyto: cytosol; Cysk: cytoskeleton; E.R: endoplasmic reticulum; Extr: extracellular; Golg: Golgi apparatus; Lyso: lysosome; Mito: mitochondria; Nucl: nuclear; Pero: peroxisome; Plas: plasma membrane; Vacu: vacuolar membrane).

a sequences were splicing errors in AnnoBeet, and were corrected by NCBI Beta vulgaris (ID 221)—Genome. These were verified by PCR amplification.

Figure 1 Multiple alignments and phylogenetic analysis of BvAQPs A. thaliana AtAQPs and D. caryophyllus DcAQPs.

Multiple alignments were performed using the default parameter of Clustal W. Phylogenetic dendrogram was generated by MEGA 6 using neighbor-joining (NJ) method with 1,000 bootstrap replicates.

Bioinformatics analysis revealed that MW of BvAQPs ranged from 25.09 to 35.08 kDa with a pI between 4.7 and 9.74 (Table 1). TIPs and SIPs were smaller (<27 kDa) than PIPs, NIPs and XIPs. TIPs were acidic while the other subfamilies were alkaline. The great majority of AQPs were predicted have six TMHs, while BvPIP1;1, BvTIP3;1, BvSIP1;1 α and BvSIP2;1 had only five TMHs and BvTIP2;2, BvNIP7;1 and BvXIP1;1 had seven TMHs.

Genes mapping on the sugar beet chromosomes, gene duplications, gene structure and alternative splicing

Twenty-eight AQP genes were unevenly mapped on eight chromosomes (Fig. 2). Seven BvAQP genes (25%) mapped on chromosome (Chr) 2, 6 BvAQP genes (21%) mapped on Chr9. Chr3, Chr5 and Chr7 each had three BvAQP genes (10%) and Chr1, Chr6 and Chr8 each had two BvAQP genes. No putative AQP was found on Chr4. Tandem duplication events were found on Chr1, Chr2, Chr5, Chr6 and Chr8 (i.e., BvPIP1;1 and BvPIP1;2, BvNIP4;1 and BvNIP4;2, BvNIP6;2 and BvNIP6;3, BvNIP5;1 and BvNIP5;2, BvPIP2;2 and BvPIP2;3).

Figure 2 Distribution of BvAQP genes on the nine B. vulgaris chromosomes.

Note: Black lines represent tandem gene duplications. Red lines represent the existence of different transcripts from single gene.

The exon-intron structures play crucial roles during plant evolution. The sugar beet AQPs exon-intron structures are showed in Fig. 3. Most PIP subfamily genes have four exons, while BvPIP1;3 had 5 and BvPIP2;4 had three. TIP subfamily genes had three exons; specifically, the third exon of BvTIP1;1 had a noncoding exon and codes a long 5′ translated region (5′ UTR). NIP subfamily genes had five exons with the exception BvNIP5;2, BvNIP6;2 and BvNIP6;3 had four exons. Most SIP subfamily genes have three exons but BvXIP1;1 only had two exons, which were similar to situations reported in common bean, tomato, potato, and so on. (Ariani & Gepts, 2015; Reuscher et al., 2013a; Venkatesh, Yu & Park, 2013). These results suggested that BvAQP gene structure is globally conserved in sugar beet. The conserved exon-intron structure provided an additional proof to support the classification results (Fig. 1).

Figure 3 BvAQP genes structure analysis.

To seek further insights into the gene structure, the splicing pattern of sugar beet BvAQP pre-mRNA sequence were analyzed. The splicing analysis revealed that BvSIP1;1 α is the result of second exon skipping (Fig. 4). The amino acid sequence of BvSIP1;1α and BvSIP1;1β shows a 74.8% similarity.

Figure 4 Alternative splicing of SIP1;1 generate two different transcripts: SIP1;1α and SIP1;1β.

Conserved resides and conserved motifs distribution

To further understand the possible physiological role and substrate specificity of BvAQPs, they were aligned and TMHs and conserved resides (NPA motifs, ar/R selectivity filter and Froger’s position) were analyzed (Table 2). The data revealed that all BvAQPs had six characteristic TMHs for AQPs, required for transport function (Fig. S1). However, BvPIP1;1 showed partial loss of TM6. The resulting missing protein domains may cause dominant negative effect on protein functions, although this need to be confirmed. All PIPs and TIPs showed the dual typical NPA motifs, but some members of NIPs such as BvNIP5;1, BvNIP5;2 and BvNIP6;1 showed that the Alanine (A) in the third residue of the second NPA motif was replace d by a Valine (V). All SIPs and XIPs showed various third residues in the first NPA motif, in which Alanine (A) was replaced by Threonine (T) or by a Leucine (L). PIPs’ ar/R selectivity filter and Froger’s position showed an apparent family-specificity compared to NPA motifs, with ar/R filter configuration typical for water-transporting AQPs (F-H-T-R) and Q/M-S-A-F-W residues in Froger’s position. TIPs contained the H-I-A/G-V/R residues in the ar/R selectivity filter and T-S/A-A-Y-W residues in Froger’s position, but BvTIP5;1 showed that the N-V-G-Y in the ar/R selectivity filter was different from other TIPs and formed a single-gene clade with the TIPs. In NIPs, the ar/R selectivity filter and Froger’s position had multiple types. BvNIP1;1, BvNIP4;1 and BvNIP4;2 showed W-V-A-R in the ar/R selectivity filter and F/L-S-A-Y-L/I in the Froger’s position. BvNIP5;1 and BvNIP5;2 showed A-I-G/A-R in the ar/R selectivity filter and F-T-A-Y-M in the Froger’s position. BvNIP6;1, BvNIP6;2 and BvNIP6;3 showed S-I-G/A-R in the ar/R selectivity filter and Y-T-A-Y-F/M/L in the Froger’s position. BvNIP7;1 showed A-V-G-R in the ar/R selectivity filter and F-S-A-Y-F in the Froger’s position. SIP1s and SIP2s showed distinct difference in the ar/R selectivity filter and the Froger’s position. BvSIP1;1 α and BvSIP1;1 β showed I/V-V-P-N ar/R selectivity filter and M-A-A-Y-W in the Froger’s position, but BvSIP2;1 showed S-N-G-S ar/R selectivity filter and F-V-A-Y-W in the Froger’s position. BvXIP1;1 showed V-S-A-R ar/R selectivity filter and F-C-A-F-W in the Froger’s position. To explore further the diversity in each group, the conserved motifs were predicted (Fig. 5). Motifs 5 and 6 appeared specifically in PIPs; Motif 8, in NIPs; Motif 9 in TIPs and XIPs.

Table 2 Conserved dual NPA motifs, ar/R (H2, H5, LE1 and LE2), Froger’s positions (P1–P5) analysis of BvAQPs.

Subfamily	Gene	NPA	Ar/R selectivity filter	Froger’s position	
		LB	LE	H2	H5	LE1	LE2	P1	P2	P3	P4	P5	
PIP	BvPIP1;1	NPA	NPA	F	H	T	R	Q	S	A	F	W	
BvPIP1;2	NPA	NPA	F	H	T	R	Q	S	A	*	*	
BvPIP1;3	NPA	NPA	F	H	T	R	Q	S	A	F	W	
BvPIP2;1	NPA	NPA	F	H	T	R	Q	S	A	F	W	
BvPIP2;2	NPA	NPA	F	H	T	R	Q	S	A	F	W	
BvPIP2;3	NPA	NPA	F	H	T	R	Q	S	A	F	W	
BvPIP2;4	NPA	NPA	F	H	T	R	M	S	A	F	W	
TIP	BvTIP1;1	NPA	NPA	H	I	A	V	T	S	A	Y	W	
BvTIP1;2	NPA	NPA	H	I	A	V	T	S	A	Y	W	
BvTIP1;3	NPA	NPA	H	I	A	V	T	S	A	Y	W	
BvTIP2;1	NPA	NPA	H	I	G	R	T	S	A	Y	W	
BvTIP2;2	NPA	NPA	H	I	G	R	T	S	A	Y	W	
BvTIP3;1	NPA	NPA	H	I	A	R	T	A	A	Y	W	
BvTIP4;1	NPA	NPA	H	I	A	R	T	S	A	Y	W	
BvTIP5;1	NPA	NPA	N	V	G	Y	T	S	A	Y	W	
NIP	BvNIP1;1	NPA	NPA	W	V	A	R	F	S	A	Y	L	
BvNIP4;1	NPA	NPA	W	V	A	R	F	S	A	Y	I	
BvNIP4;2	NPS	NPA	W	A	A	R	L	S	A	Y	I	
BvNIP5;1	NPS	NPV	A	I	G	R	F	T	A	Y	M	
BvNIP5;2	NPS	NPV	A	I	A	R	F	T	A	Y	M	
BvNIP6;1	NPS	NPV	S	I	G	R	F	T	A	Y	F	
BvNIP6;2	NPA	NPA	S	I	G	R	Y	T	A	Y	M	
BvNIP6;3	NPA	NPA	S	I	A	R	Y	T	A	Y	L	
BvNIP7;1	NPA	NPA	A	V	G	R	F	S	A	Y	F	
SIP	BvSIP1;1α	NPT	NPA	I	V	P	N	M	A	A	Y	W	
BvSIP1;1β	NPT	NPA	V	V	P	N	M	A	A	Y	W	
BvSIP2;1	NPL	NPA	S	N	G	S	F	V	A	Y	W	
XIP	BvXIP1;1	NPT	NPA	V	S	A	R	F	C	A	F	W	

Figure 5 BvAQPs protein motifs identified by MEME.

Using the complete amino acid sequences of BvAQPs. Combined p-values are indicated in (A) and different motifs were shown by different colors and numbered from 1 to 10 in (B).

Subcellular localization

Plant-mPLoc prediction was used to predict the BvAQPs subcellular localization (Table 1). PIPs, NIPs, TIPs and SIPs (except BvSIP1;1 β) were predicted to localize to plasma membrane and all TIPs were also predicted to localize on vacuolar membrane. However, subcellular localizations predicted by WoLF PSORT were diverse and not always in agreement with that predicted by Plant-mPLoc (Table 1). For example, most PIPs were predicted to localize on plasma membrane. TIPs were predicted to localize on vacuolar membrane, plasma membrane as well as in the nucleus, mitochondria, cytosol and chloroplast. NIPs, SIPs and XIPs were predicted to localize on vacuolar membrane and plasma membrane. Although these predications represent a starting base, more experiments, i.e., with tagged proteins, are will be useful better characterize the subcellular localization of BvAQPs.

Figure 6 Expression profiles of the 28 BvAQP genes in different plant organs and tissues.

Figure 7 Expression pattern of the 23 BvAQP genes in young leaf under abiotic stress.

Expression analysis of sugar beet AQP genes in different organs and in response to salt & heat stress

We used the available RNA-seq data from different organs and young leaf under heat, salt stress treatment of B. vualgaris (Dohm et al., 2014; Minoche et al., 2015) to evalute tha expression of BvAQPs. 27 BvAQP genes (96%) were expressed in at least one tissue (Fig. 6, Table S1) and one BvAQP gene (NIP6;3) did not show any expression in all tested plant organs. Eighteen BvAQP genes (64%) expressed in all organs, including six PIP, four TIP, four NIP, three SIP and one XIP. Twenty-six BvAQP genes (92%) expressed in taproot, with BvPIP1;3, BvTIP1;1 and BvPIP2;4 exhibiting the highest expression levels. Twenty-six AQP genes (92%) were expressed in inflorescence, among which BvPIP2;4, BvPIP1;3 and BvTIP1;1 exhibiting the highest expression levels. Twenty-two AQP genes (79%) were expressed in seed (with BvTIP3;1, BvPIP2;1, BvPIP1;3 and BvTIP1;1 showing the highest expression levels). Only 18 AQP genes (64%) were expressed in leaf (BvTIP1;1, BvPIP1;3 and BvTIP2;1 showed the highest expression levels). Most genes were expressed constitutively in the tested tissues, while one gene (BvTIP5;1) showed tissue-specific expression in the inflorescence. In addition, the transcripts of BvTIP1;2 and BvNIP7;1 were also shown to be relatively abundant in inflorescence and BvTIP2;2 was highly expressed in taproot and seeding.

The expression patterns of sugar beet AQP genes were found to fluctuate under different stress conditions. Eleven AQP genes (44%) expression were down-regulated both under salt and heated stress. These include BvPIP1;1, BvPIP1;2, BvPIP2;3, BvNIP4;1, BvNIP4;2, BvNIP5;2, BvNIP6;1, BvNIP7;1, BvSIP1;1 α, BvSIP1;1 β and BvSIP2;1 (Fig. 7). In contrast, the expression of BvPIP2;4, BvTIP1;1 and BvTIP2;1 (12%) were up-regulated both under salt and heated stress. Interestingly, nine AQP genes showed different expression in response to salt or heat treatments. Eight among these AQP genes (32%; BvPIP1;3, BvPIP2;1, BvPIP2;2, BvTIP1;3, BvTIP4;1, BvNIP1;1, BvNIP5;1 and BvXIP1; 1) were up-regulated under heat treatment but down-regulated under salt treatment. Notably, members of PIP genes and XIP genes (BvPIP1;3, BvPIP2;1, BvPIP2;2, BvXIP1;1) were highly up-regulated under heat stress but slightly down-regulated under salt stress. Only BvTIP2;2 (4%) was up-regulated under salt but down-regulated under heat stress.

Interestingly, BvNIP6;3 wasn’t expressed in all organs, neither in young leaf under salt and heat stress (Fig. 6; Table S1), and BvNIP6;3 formed a single-gene clade within the NIP6s, with the highest MV (35.08kDa) and longest protein length (Table 1). In addition, gene duplication analysis revealed BvNIP6;3 and BvNIP6;2 were tandem duplication (Fig. 2) and suggested that BvNIP6;3 was originated from BvNIP6;2 and was pseudo-genes.

Other genes, such as BvTIP1;2, BvTIP3;1, BvNIP6;2 and BvNIP6;3 were not expressed in non-treated or in stressed leafs. BvTIP5;1 was not expressed in control or heat stressed young leafs, but showed only very expression in young leaf under salt stress (Table S1).

Discussion

Many studies have confirmed that AQP genes were involved in plant water transport, in regulating growth and development and that they are widely distributed in animals and plants. Most functional studies of AQP genes were mainly conducted using model plants, such as A. thaliana, N. tabacum L., Z. mays L. and O. sativa L. Almost nothing or very little is known about B. vulgaris AQPs. Here we report the first investigation of AQPs in sugar beet. Twenty-eight AQP genes were identified, including seven PIPs, eight TIPs, nine NIPs, three SIPs and one XIP.

The number of AQPs genes in B. vulgaris is similar to that of D. caryophyllus (Morita et al., 2017), a closely related species belonging to the same family. B. vulgaris contains less AQP genes compared to plants that did undergo 1-3 times of whole genome duplication (WGD), fragment duplication (FD) and/or tandem duplication (TD), such as G. max L (Zhang et al., 2013), Brassica rapa (Tao et al., 2014), Z. mays L. (Chaumont et al., 2001), O. sativa L (Nguyen, Moon & Jung, 2013) and P. trichocarpa (Gupta & R, 2009). B. Vulgaris contains more AQP genes than basal plants such as P. patens, S. moellendorffii (Bowers et al., 2003; Doyle et al., 2008; Gupta & R, 2009; Yu & Yang, 2002; Zhang et al., 2013). Almost all subgroups of PIPs, TIPs, SIPs and XIPs are present in the Caryophyllaceae plants B. vulgaris and D. caryophyllus (Morita et al., 2017).

The relatively small number of AQP genes in sugar beet, compared to other higher plant species, is likely due to the fact that B. vulgaris did not undergo WGD. Moreover, the rapid expansion of NIP subfamily which led to the production of new subgroups, might have occurred after the divergence of the basal core dicot and core dicot. The divergence and proliferation of NIP subfamily may be an adaptive response to an ever-changing environment, playing crucial role in plants disease resistance and multi-stress (Liu & Zhu, 2010). Some studies also suggested that NIPs allow larger solutes, such as silicic acid, to permeate. O. sativa L genes OsNIP2;1 (Lsi1), OsNIP2;2 (Lsi6), and Hordeum vulgare L. HvNIP2;1 (HvLsi1) transport silicon across the biomembrane and enhance the resistance of plants to biotic and abiotic stress (Chiba et al., 2009; Ma et al., 2006; Yamaji & Ma, 2009; Yamaji, Mitatni & Ma, 2008), and O. sativa L OsNIP2;1 could also be permeable to water, urea, boric acid, arsenite (Ma et al., 2008; Mitani, Yamaji & Ma, 2008).

To analyze the evolutionary relationship and BvAQPs putative functions, an unrooted phylogenetic tree was constructed using aquaporins from A. thaliana (in which a complete set of AtAQP genes is well known) and D. caryophyllus, closely related species to B. vulgaris (Johanson & Kjellbom, 2001; Morita et al., 2017). During the process of cluster analysis, two carnation AQPs (DcSIP2;1 and DcNIP5;1) aroused our attention and made us confused in classification at once. Previously, the classification of DcSIP2;1 and DcNIP5;1 was not well resolved (Morita et al., 2017). In the case of DcSIP2;1, it shared the highest similarity of 67% with BvSIP1;1 and 51% with AtSIP1;1, and also shared the same NPA motifs, Ar/R selectivity filter and Froger’s position with BvSIP1;1. It clustered closer to the SIP1 subgroup. In the case of DcNIP5;1, it farther clustered to the NIP5 subgroup and clustered closer to the NIP6 subgroup. The closest homolog of DcNIP5;1 is BvNIP6;3 (similarity is 51%) and both proteins share the same Ar/R selectivity filter (S-I-A-R) with BvNIP6;3 and BvNIP6;1, with a similar Ar/R selectivity filter (A-I-A-R) to DcNIP6;1. Therefore, DcNIP5;1 belongs to NIP6 subgroup.

Alternative splicing is a mechanism by which genes can produce multiple transcript variants protein products, which allows in turn to increase the diversity of gene functions. Alternative splicing plays an important role in various processes such as development, response to pathogen and to various abiotic stresses (Bove et al., 2008; Gassmann, 2008; Jang et al., 2009; Reddy & Golovkin, 2010). A pair of splice variants (BvSIP1;1 α and BvSIP1;1 β) were found in BvAQPs. BvSIP1;1 α resulted from second exon skipping. First exon and second exon encoded the same ‘NPT’ type NPA motif, so BvSIP1;1 α and BvSIP1;1 β shared similar amino acid sequence and protein topology. These two genes had a similar expression pattern in different organs. It is possible that BvSIP1;1 α and BvSIP1;1 β may have redundant function. Exon skipping splicing were also found in S. tuberosum, while St-SIP1;1 α and St-SIP1;1 β also resulted from second exon skipping and shared similar expression pattern (Venkatesh, Yu & Park, 2013). However, the identified alternative splicing events found for BvAQPs genes were not reported in other plant species, suggesting that selective splicing may be species-specific.

NPA motifs, ar/R selectivity filter and Froger’s position were reported involved in substrate selection and transport activity. BvAQPs functions could be conferred based on the comparison of these residues with other plants AQPs. BvPIPs showed typical NPA motif, F-H-T-R highly conserved ar/R selectivity filter and Q-S-A-F-W Froger’s position, this composition is highly conserved in PIPs of A. thaliana (Johanson & Kjellbom, 2001), Z. mays (Chaumont et al., 2001), S. lycopersicum (Reuscher et al., 2013b), P. trichocarpa (Gupta & R, 2009), G. max (Zhang et al., 2013) and B. rapa (Tao et al., 2014). This composition of PIPs was reported to likely regulate root and leaf hydraulics, facilitate the CO2 diffusion, affect photosynthesis (Gupta & R, 2009). Therefore, B. vulgaris homologous PIPs may have similar roles in regulating water absorption, plant hydraulics and CO2 diffusion. Based on the SDPs analysis proposed by Hove and Bhave (Hove & Bhave, 2011) (Table 3; Fig. S2), all B. vulgaris PIPs represented urea-type SDPs, BvPIP1;1 and BvPIP1;2 represented boric acid-type SDPs, all PIP2s represented H2O2-type SDPs, thus supporting conserved functions. In addition, BvPIP1;2, BvPIP1;3 and BvPIP2;1 seemed to represent novel CO2-type SDPs (I/L/V-M-C-A-I/V-D/H/K-W-D-W) with the substitution of Ile for Met in SDP2 and the substitution of D to H/k in SDP6. Although the ar/R selectivity filter varied highly, plant TIPs were shown to transport water as efficiently as PIPs (Zhi et al., 2015). Additionally, BvTIP1;1, BvTIP1;2, BvTIP1;3, BvTIP3;1 and BvTIP4;1 contained dual conserved NPA motifs, H-I-A-V/R ar/R selectivity filter and T-S-A-Y-W Froger’s position, which had the same composition of Citrus sinensis AQPs, such as CsTIP1s, CsTIP3s showed to transport urea and H2O2 (Hove & Bhave, 2011). Compared to other TIPs, BvTIP2;1 and BvTIP2;2 had H-I-G-R ar/R selectivity filter, and this type ar/R selectivity filter was experimentally proven to transport formamide (Hove & Bhave, 2011), suggesting BvTIPs play a crucial role in transporting a wide range molecules. All TIPs (exception of BvTIP5;1) represented urea-type SDPs, indicting transporting urea function. BvTIP1;1 and BvTIP3;1 represented H2O2-type SDPs, indicting transporting H2O2 potential capacity.

Substrate selection specificity of NIPs is largely determined by two pores formed by NPA motifs and ar/R selectivity filter respectively. ‘W-V-A-R’ type ar/R selectivity filter were proven to be permeable to uncharged molecules i.e., glycerol, formamide and water, implying that BvNIP1;1 and BvNIP4;1 may have similar functions. ‘A/S/T-V/I-A/G-R’ type ar/R selectivity filter were reported can transport glycerol, formamide and larger solutes, like urea, boric, but were impermeable to water. Besides, NPS/NPV aqueous pore and A-I-G-R ar/R selectivity filter was known as a boric acid transporter in A. thaliana AtNIP5;1, and orthologs BvNIP5;1 can be involved in boron transport in B. vulgaris, suggesting that NIPs could be involved in the transport of larger solutes. Based on SDPs analysis, all NIPs (except BvNIP7;1) are potential urea transporters, while BvNIP4;2 is a potential H2O2 transporters, BvNIP5;2 and BvNIP6;1 are potential boric acid transporters and finaly BvNIP1;1 could represent a novel NH3-type SDPs with H replaced K/L/N/V at SDP2. It should be noticed that we were unable to find NIP2 subgroup and silicic acid-type SDP AQPs in sugar beet, suggesting that other types of SDPs AQPs could be involved in absorption of silicic during plant growth and development, in sugar beet. Sugar beet SIPs and XIPs exhibited great variety in ar/R selectivity filter and Froger’s position, with a variable first NPA motif compared to other AQPs subfamilies. It has been reported that A. thaliana SIPs (AtSIP1;1 and AtSIP1;2) localize to endoplasmic reticulum (ER) and facilitate water transport. SDPs analysis suggest that BvXIP1;1 can transport boric acid, it is thus possible that BvXIP has different role in B. vulgaris.

Table 3 Identified typical SDPs in BvAQPs.

Aquaporin	Specificity-determining positions	
		SDP1	SDP2	SDP3	SDP4	SDP5	SDP6	SDP7	SDP8	SDP9	
Typical NH3 transporter	F/T	K/L/N/V	F/T	V/L/T	A	D/S	A/H/L	E/P/S	A/R/T	
	BvNIP1;1	F	H	F	T	A	D	L	E	T	
Typical Boric Acid transporter	T/V	I/V	H/I	P	E	I/L	I/L/T	A/T	A/G/K/P	
	BvPIP1;2	T	I	H	P	E	L	L	T	P	
	BvPIP1;3	T	I	H	P	E	L	L	T	P	
	BvNIP5;2	T	I	H	P	E	L	L	A	P	
	BvNIP6;1	T	I	H	P	E	L	L	A	P	
	BvXIP1;1	T	I	H	P	E	L	L	T	P	
Typical CO2 transporter	I/L/V	I	C	A	I/V	D	W	D	W	
	BvPIP1;2	V	M	C	A	I	D	W	H	W	
	BvPIP1;3	V	M	C	A	I	H	W	D	W	
	BvPIP2;1	I	M	C	A	V	K	W	D	W	
Typical H2O2 transporters	A/S	A/G	L/V	A/F/L/T/V	I/L/V	H/I/L/Q	F/Y	A/V	P	
	BvPIP2;1	A	G	V	F	I	H	F	V	P	
	BvPIP2;2	A	G	V	F	I	H	F	V	P	
	BvPIP2;3	A	G	V	F	I	H	F	V	P	
	BvPIP2;4	A	G	V	F	I	H	F	V	P	
	BvTIP1;1	S	A	L	A	I	H	Y	V	P	
	BvTIP3;1	A	A	L	T	I	H	Y	V	P	
	BvNIP4;2	S	A	L	L	I	L	Y	V	P	
Typical silicic acid transporters	C/S	F/Y	A/E/L	H/R/Y	G	K/N/T	R	E/S/T	A/K/P/T	
	Not found										
Typical urea transporters	H	P	F/I/L/T	A/C/F/L	L/M	A/G/P	G/S	G/S	N	
	BvPIP1;1	H	P	F	L	L	P	G	G	N	
	BvPIP1;2	H	P	F	L	L	P	G	G	N	
	BvPIP1;3	H	P	L	F	L	P	G	G	N	
	BvPIP2;1	H	P	F	L	L	P	G	G	N	
	BvPIP2;2	H	P	F	F	L	P	G	G	N	
	BvPIP2;3	H	P	F	F	L	P	G	G	N	
	BvPIP2;4	H	P	F	F	L	P	G	G	N	
	BvTIP1;1	H	P	F	F	L	A	G	S	N	
	BvTIP1;2	H	P	F	F	L	P	G	S	N	
	BvTIP1;3	H	P	L	F	L	A	G	S	N	
	BvTIP2;1	H	P	F	A	L	P	G	S	N	
	BvTIP2;2	H	P	F	A	L	P	G	S	N	
	BvTIP3;1	H	P	F	L	L	P	G	S	N	
	BvTIP4;1	H	P	L	A	L	L	G	S	N	
	BvNIP1;1	H	P	I	A	L	P	G	S	N	
	BvNIP4;1	H	P	L	A	L	P	G	S	N	
	BvNIP4;2	H	P	I	A	L	T	G	S	N	
	BvNIP5;1	H	P	I	A	L	P	G	S	N	
	BvNIP5;2	H	P	I	A	L	P	G	S	N	
	BvNIP6;1	H	P	I	A	L	P	G	S	N	
	BvNIP6;2	H	P	L	A	L	P	G	S	N	
	BvNIP6;3	H	P	I	A	L	P	G	S	N	

Sequence alignment revealed that BvPIP1;1 codes for a truncated protein that lacks TM6 domain. The absence of TM6 domain possibly affects subcellular localization, proper folding, and oligomerization and transport activity of this truncated AQP in B. vulgaris. Similarly truncated AQP TdPIP2;1 has been reported to affect water transport activity in wheat (He et al., 2012; Hu, 2012). RNA-seq analysis revealed that AQP genes were expressed in all examined tissues of B. vulgaris, thus similar to previously reported data in maize (Chaumont et al., 2001), A. Arabidopsis (Quigley, 2001), tomato (Reuscher et al., 2013a) and potato (Venkatesh, Yu & Park, 2013). Total transcript abundance in the different examined tissues showed high expression levels in organs involved in water absorption, transport and evaporation, such as taproot and leaf, suggesting the main role of AQPs in water and nutrients transport. The RNA-seq data suggest that BvAQPs may play a key role in radial and axial water transportation in sugar beet, thus in agreement with Amodeo et al. (1999). Many AQPs show similar expression patterns, suggesting that they may act synergistically in some tissues. Co-expression of tandem PIP2-PIP1 dimers in Xenopus oocytes showed that they can form PIP2-PIP1 hetero-tetramers and synergistically increase water transport capacity (Bellati et al., 2010; Jozefkowicz et al., 2013; Jozefkowicz et al., 2015). BvPIP1;3, BvTIP1;1 and BvPIP2;4 were considerably abundant in taproot and are likely involved in water and nutrients transport in root. BvPIP2;4, BvPIP1;3 and BvTIP1;1 were abundant in inflorescence and seeding, and could be involved in monitoring the water balance in young tissues. According to their expression profiles, BvTIP1;1, BvPIP1;3 and BvTIP2;1 could play key roles in leaf hydraulics. BvTIP3;1, BvPIP2;1, BvPIP1;3 and BvTIP1;1 were considerably abundant in seed and therefore could be involved in water balance in seed. BvPIP1;3 and BvTIP1;1 show high expression levels in all examined tissues, while BvTIP3;1 were only expressed in abundance in seed, and showed very low expression in other organs, suggesting a key role in seed development. BvTIP3;1 closest homolog Arabidopsis AtTIP3;1 and castor bean RcTIP3;1 also shared similar expression pattern. AtTIP3;1 was reported to be seed- and embryo-specific AQP (Johnson, Herman & Chrispeels, 1989). RcTIP3;1 expressed preferentially in endosperm of developing seeds and considerably low in germinating seed (Zou et al., 2015). Moreover, BvPIP2;2 and BvTIP2;2 were highly expressed in seeding where they may control the water balance. In total, several AQP genes (like BvTIP3;1, BvPIP2;1, and BvPIP1;3) play a constitutive role and some AQP genes (like BvTIP3;1, BvPIP2;2 and BvTIP2;2) play a specialized function in specific plants organs. In addition, it is noteworthy that several putative non-aqua transporter-encoding genes (i.e., BvNIP7;1, BvNIP4;1, BvXIP1;1, BvNIP5;2, BvNIP6;1 and BvSIP2;1) were shown to be relatively high abundant in certain tissues. Compared to others tissues, transcript level of BvNIP4;1, BvNIP5;2 and BvNIP6;1 was considerably high in taproot. Similar to BvNIP7;1, Arabidopsis AtNIP7;1 (specifically expressed in anthers) encode for a less efficient boric acid transporter, compared to AtNIP5;1 and AtNIP6;1. It is possible that BvNIP7;1 plays a role in B absorption and balance in inflorescence tissue. BvXIP1;1 had relatively high expression level in leaf. Similarly, BvXIP1;1 the closest homolog to RcXIP1;1, also showed high expression levels in leaf, but their exact functions remain unclear and further functional investigation are required.

In this study, a large number of BvAQP genes showed transcriptional changes when exposed to salt and heat stresses. For example, BvTIP1;1 and BvTIP2;1 showed strong induction (Log2-based value >1) after salt stress; BvPIP2;2, BvTIP1;1, BvTIP2;1 and BvXIP1;1 showed strong induction (Log2-based value >1) after heat stress, suggesting an extensive response of BvAQP genes to abiotic stress and a potential function for improving sugar beet resistance to abiotic stress. The results were consistent with previously reported data that showed that BvPIP2;2 expression was up-regulated in response to high salt stress (Skorupakłaput et al., 2015). Biochemical and genetic evidence has demonstrated that some AQP genes (like TaAQP7, TaAQP8, TaNIP, MaPIP1;1, MusaPIP2;6 and NtAQP1) improve plants resistance to abiotic stress (Gao et al., 2010; Hu, 2012; Sade, 2010; Sreedharan, Shekhawat & Ganapathi, 2015; Xu et al., 2014; Zhou et al., 2012). Additionally, we noticed that BvTIP1;1 and BvTIP2;1 were strongly induced by both salt and heat stresses, suggesting that these two AQPs may play key roles in B. vulgaris adaption to environmental changes, e.g., heat and salt stresses.

Conclusions

Twenty-eight BvAQP genes were identified based on genome data, chromosome distribution, phylogenetic, protein characteristics, gene structural, gene duplication, conserved motifs and RNA-seq expression analysis of BvAQP genes were further researched. The results suggested that AQPs had multiple functions in different tissues; several BvAQP genes responded to abiotic stresses and improved the plants’ resistance to abiotic stresses. The potential functions of BvAQPs were predicted and discussed based on NPA motifs, ar/R selectivity filter, Froger’s positions and SDP positions analysis. This study provide a useful resource for identifying and characterizing BvAQPs and a base for the breeding and genetic engineering of sugar beet.

Supplemental Information

Fig. S1 Alignment of deduced amino acid sequences of BvAQPs

Click here for additional data file.

Fig. S2 SDPs position of AQPs

Click here for additional data file.

Table S1 Transcription abundance of BvAQP genes

Click here for additional data file.

The authors appreciate those contributors who make the sugar beet genome and transcriptome data accessible in public databases.

Additional Information and Declarations

Competing Interests

Author Contributions

Data Availability

The authors declare there are no competing interests.

Weilong Kong conceived and designed the experiments, performed the experiments, analyzed the data, wrote the paper, prepared figures and/or tables, reviewed drafts of the paper.

Shaozong Yang and Yulu Wang performed the experiments.

Mohammed Bendahmane and Xiaopeng Fu conceived and designed the experiments, contributed reagents/materials/analysis tools.

The following information was supplied regarding data availability:

The raw data has been supplied as Supplementary Files.

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
