# Peer review of "Genome-wide identification and characterization of aquaporin gene family in Beta vulgaris"

_PeerJ, doi:10.7717/peerj.3747_

## Round 0.1 · original submission · Major Revisions

I have read the manuscript and both reviews and agree that the manuscript needs significant revision before it is ready to be published. I want to emphasize that the manuscript needs significant improvement in the English. I highly suggest the authors have it professionally edited to improve the language. In its current form it is not publishable.

Please address all of the reviewer’s comments.

Specifically the authors need to better explain why an E-value of e-5 was used to identify homolog from a genome. The authors need to either provide more evidence for analysis that are based solely on automatic genome annotation or discuss how these results could be misleading. Since this manuscript is based on data in databases for analysis, I agree with reviewer two that additional relevant papers need to be discussed in the discussion. Please ensure that all figures and tables are relevant to the manuscript. The discussion on RNA-seq data in different conditions should be minimized as suggested by reviewer two. I agree that few conclusions can be made when all these experiments were conducted under different experimental conditions with no common controls.

Overall, the manuscript needs significant work but it does present valid findings and could be appropriate for PeerJ given a significant revision.

Reviewer 1 ·

Basic reporting

The study presents the genome-wide identification of aquaporin (AQP) genes in sugar beet (Beta vulgaris). However, the English language of the manuscript should be improved to ensure that your international audience can clearly understand, and I highly recommend the authors to invite a native English speaking colleague to review the manuscript.

Experimental design

In this manuscript, the HMM and BLASTP search were adopted to identify BvAQPs. However, in most cases, the TBLASTN search with the E-value of e-5 was used to identify homolog from a certain genome. Please explain why?

Validity of the findings

Several analyses (e.g. gene structure and motif analysis) presented in this study were purely based on the automatic genome annotation. I am afraid it would be misleading. More evidence should be provided.

Additional comments

1. The title of the manuscript is hard to understand and should be changed.
2. Once the abbreviation of a certain term was described, the abbreviated form is enough in the following scenes, e.g., aquaporin (AQP) in line 15.
3. The statement in lines 54-56 is not accurate, because XIPs have been widely found in dicots, mainly excluding Brasslcaceae plants.
4. The comparison of AQPs in sugar beet and carnation were described, however, more details of carnation AQPs should be introduced in the part of “Introduction”.
5. The HMM and BLASTp search were adopted to identify BvAQPs. However, I still not sure the gene family is complete, especially when the automatic genome annotation was used.
6. Analysis of the gene structures purely based on the automatic annotation would be misleading.
7. Sequence alignments presented in Fig_S2 should be manually checked.

Reviewer 2 ·

Basic reporting

English language should be significantly improved throughout the entire paper, possibly with the assistance of a native english-speaking person.
Literature references seem adequate, with the relevant exception of the part of the paper dealing with functional aspects; this is especially true considering that the authors discuss the relevance of the AQPs they bioinformatically characterized, and their possible behaviour in conditions of water or salt stress, but they fail to quote the few relevant papers on this very same issue (acquaporins in sugar beet), like Amodeo et al., 1999, J.Exp. Botany 50, 509-516; or Skorupa-Klaput et al., 2015, Biologia 70/4, 467-477. I believe that these papers need to be taken into consideration during the discussion, especially because the authors did not made their own stress experiments measuring AQPs expression levels, but only relied on RNA-seq data from public datasets. Had the authors dealt only with a bioinformatic analysis of the AQPs gene structure, this problem would not have been so evident.
Figures and tables are sufficiently clear but not sufficiently informative, as in many cases (e.g. figure 4) an explanation guiding the reader in the examination of the figure would be advisable.
In the text there are too many lists of items which make reading quite difficult; examples of this are in lines 222-232 or 250-271, where the same data presented in tables is listed.
Results are relevant only as far as the analysis of the structure of sugar beet AQPs is concerned. This is in my opinion the core of the work, and the only informative and interesting part of the paper. My suggestion to the authors is to eliminate the part in which they discuss RNA.seq data in different conditions; I would say that only making original experiments and not data mining, it is possible to draw meaningful conclusions, though this is my personal opinion. As an alternative, the authors should submit the paper to a bioinformatics journal. I also suggest to change the title, making much clearer to the reader that in this paper only bioinformatics will be found.

Experimental design

I think the authors explored adequately the most relevant information present on different databases, using a sufficient number of databases and softwares so to extract as much information as possible from already existing data. No original sequencing work was carried out, but the comparisons provide adequate analysis of the state of the art of AQPs sequence heterogeneity and structure in sugar beet.
Methods, all bioinformatic in nature, are sufficiently described.

Validity of the findings

Rather than the true novelty of the results, we should consider that all data used to make the work and draw the conclusions is from the databases.
This is indeed the first report comparing AQPs sequences in sugar beet, and this is the main character of novelty, as also the authors state, allowing comparison with other plant species of the AQPs structure and types. However, sequences of the strictly sugar beet-related subspecies Beta vulgaris ssp maritima appear not to have been included in the analysis, or at least they are never explicitly mentioned. I think that an effort in this direction should be made, especially due to the high relevance that this subspecies had for sugar beet breeding and for its characteristics of resistance to drought and salt stress. In case no AQPs-like sequences are present in the database for Beta vulgaris ssp. maritima, this should be clearly indicated.

Additional comments

In this paper, unfortunately the sugar beet plant does not "emerge" in its characteristics, as you completely focused on the structural features of AQPs, never trying to link them to a knowledge of this important crop and of its wild relatives.

---

## Round 0.2 · accepted · Accept

I think the authors have made the necessary revisions and this manuscript should be accepted for publication.

Reviewer 1 ·

Basic reporting

The English language of the manuscript should be further improved to ensure that your international audience can clearly understand.

Experimental design

The genome-wide identification should base on the genome but not the annotated proteome, independent of the quality and accuracy of a certain genome.

Validity of the findings

No.

Additional comments

1. The title of the manuscript is suggested to be “Genome-wide identification and characterization of the aquaporin gene family in Beta vulgaris”.
2. XIPs have been widely found in dicots, mainly excluding Brasslcaceae plants.
3. The English language of the manuscript should be improved before publication.